# The Impact of Urban Renewal on Land Surface Temperature Changes: A Case Study in the Main City of Guangzhou, China

**Zhi Qiao** [1], **Luo Liu** [2], **Yuanwei Qin** [3], **Xinliang Xu** [4], **Binwu Wang** [5] and **Zhenjie Liu** [6,*]

1    Key Laboratory of Indoor Air Environment Quality Control, School of Environmental Science and Engineering, Tianjin University, Tianjin 300350, China; qiaozhi@tju.edu.cn
2    Guangdong Province Key Laboratory for Land use and consolidation, South China Agricultural University, Guangzhou 510642, China; liuluo@scau.edu.cn
3    Department of Microbiology and Plant Biology, Center for Spatial Analysis, University of Oklahoma, Norman, OK 73019, USA; yuanwei.qin@ou.edu
4    State Key Laboratory of Resources and Environmental Information Systems, Institute of Geographical Sciences and Natural Resources Research, Chinese Academy of Sciences, Beijing 100101, China; xuxl@lreis.ac.cn
5    Policy Research Center, Ministry of Housing and Urban-Rural Development of the People's Republic of China, Beijing 100835, China; wangbw@mohurd.gov.cn
6    Guangdong Provincial Key Laboratory of Urbanization and Geo-simulation, School of Geography and Planning, Sun Yat-sen University, Guangzhou 510275, China
*    Correspondence: liuzhj66@mail2.sysu.edu.cn; Tel.: +86-20-85288307

**Abstract:** To improve land use efficiency, urban renewal must also consider urban microclimates and heat islands. Existing research has depended on manual interpretation of high-resolution optical satellite imagery to resolve land surface temperature (LST) changes caused by urban renewal; however, the acquired ground time series data tend to be uneven and unique to specific frameworks. The objective of this study was to establish a more general framework to study LST changes caused by urban renewal using multi-source remote sensing data. Specifically, urban renewal areas during 2007–2017 were obtained by integrating Landsat and yearly Phased Array type L-band Synthetic Aperture Radar (PALSAR) images, and LST was retrieved from Landsat thermal infrared data using the generalized single-channel algorithm. Our results showed that urban renewal land (URL) area accounted for 1.88% of urban land area. Relative LST between URL and general urban land (GUL) of Liwan, Yuexiu, Haizhu, and Tianhe districts dropped by 0.88, 0.42, 0.43, and 0.10 K, respectively, whereas those of Baiyun, Huangpu, Panyu, and Luogang districts presented opposite characteristics, with a rise in the LST of 0.98, 1.03, 1.63, and 2.11 K, respectively. These results are attributable to population density, building density, and landscape pattern changes during the urban renewal process.

**Keywords:** urban renewal; land surface temperature; remote sensing; urban heat island

## 1. Introduction

Human-oriented urbanization and related land-use change processes significantly affect the thermal environment of cities and their surrounding areas by transforming the natural landscape into an impervious surface [1–3]. The most prominent urban thermal environment problem is the development of urban heat islands (UHIs). As one of the most obvious characteristics of urban climate change caused by construction and human activity, UHIs constitute heat accumulation in urban areas due to their higher surface temperature compared with the surrounding suburbs and rural regions [4,5].

By changing the structure and function of the ecosystem and terrestrial surface energy exchange processes, UHIs proceed to negatively affect the urban climate and urban hydrology [6,7]. UHIs also seriously threaten the suitability of urban living environments and the health of residents [8–10]. With the continuing advance of urbanization, the scale and intensity of UHIs will become more and more serious. Therefore, it is of practical importance to understand how urbanization influences the UHI effect and microclimates, for future planning, ecological protection, and sustainable development in these areas.

While urbanization can promote developmental changes that improve people's living standards, it can also lead to increasingly serious UHI effects. To understand more fully the mechanism and characteristics of urbanization on UHIs, many studies have analyzed the relationship between urbanization and land surface temperature (LST) [11–16]. These related studies have focused on the spatiotemporal characteristics and influencing factors of UHIs [17–20]. For example, based on MODIS 8-day composite LST products, the differences in surface urban heat island intensity (SUHII) among seasons and cities, and between day and night, were analyzed in the Yangtze River Delta urban agglomeration; the specific impact of climatic factors (e.g., precipitation, air temperature, solar radiation, and wind speed) and urbanization factors (build-up intensity, population density, and urban area size) on SUHII were further analyzed using Pearson's correlation coefficients and stepwise linear regression [21]. Additionally, there has been great interest in simulations of UHI spatial patterns [22,23] and the interaction between UHIs and the landscape pattern under urbanization [24–26]. Most of the existing research has concentrated on and is limited to the impact of urban expansion on UHIs. However, as an important component of urbanization, few studies have considered how urban renewal influences UHIs and the urban microclimate.

Inefficient urban land use and insufficient land reserves are inevitable with urbanization. As such, rapid urban sprawl tends to accompany urban renewal [27,28], i.e., the demolition and/or renovation of old industrial, commercial, and residential areas and urban villages, to improve the vitality of the area and its relationship with the surrounding environment [29,30]. For example, some urban villages, old factories, and polluting facilities can be rebuilt as residential, office, commercial, or leisure complexes. Abandoned houses and shanty towns may be modernized or removed to allow space for public facilities, such as green parks, markets, and parking lots. As urban renewal can be beneficial to land use efficiency and the environment by changing urban morphology and development patterns, it has gradually become a key focus in urban planning and urban sustainable development management [31].

However, research on the influence of urban renewal on LST has been limited, with few studies published to date [32–34]. For example, based on high-resolution Worldview images and multi-temporal Advanced Spaceborne Thermal Emission and Reflection Radiometer (ASTER) thermal infrared images, manually extracted urban renewal areas and LST change in the region have been investigated over different time periods [33]; however, it is difficult to apply the analysis results widely as they typically depend on manual interpretation of optical satellite imagery, which is prone to adverse atmospheric conditions and cloud pollution [15,35], and the expense of high-resolution Worldview images limits the wide usage of this method. Consequently, the resulting ground time series data may not be generally acquired and provide a good representation of actual conditions in different regions of interest.

Therefore, we established a more general framework to study LST changes caused by urban renewal integrating free and open dataset including optical images, radar images, and thermal infrared images. The main objectives of this study were: (1) to interpret automatically and obtain city urban renewal areas according to the integration of Landsat images and Advanced Land Observation Satellite (ALOS) Phased Array type L-band Synthetic Aperture Radar (PALSAR) images; (2) to retrieve the LST from Landsat thermal infrared data over different periods; and (3) to analyze the impact of urban renewal on LST changes to provide a new perspective on the impact of urban renewal on microclimates.

## 2. Materials and Methods

### 2.1. Study Area

As the most important political, economic, cultural, and technological center in southern China, Guangzhou City has undergone rapid economic development and urbanization since its opening and reform [36]. With large-scale human and land development activity, the urban landscape and climate have experienced significant changes, putting Guangzhou City at great environmental risk from thermal changes [37]. The study area includes a core region, consisting of Liwan, Yuexiu, Haizhu, and Tianhe districts, and a peripheral region that includes Baiyun, Huangpu, Panyu, and Luogang districts (Figure 1). According to the statistics, the land area of the core region is about 279.63 km$^2$, and its urban population and gross domestic product (GDP) in 2017 were 5.47 million and 1033.73 billion, respectively. The land area of the peripheral region is about 1809.90 km$^2$, and its urban population and GDP in 2017 were 4.61 million and 702.99 billion, respectively. The combined effect of accelerated urbanization and insufficient land resources has led to rapid urban renewal processes, resulting in microclimate and thermal environmental changes [33].

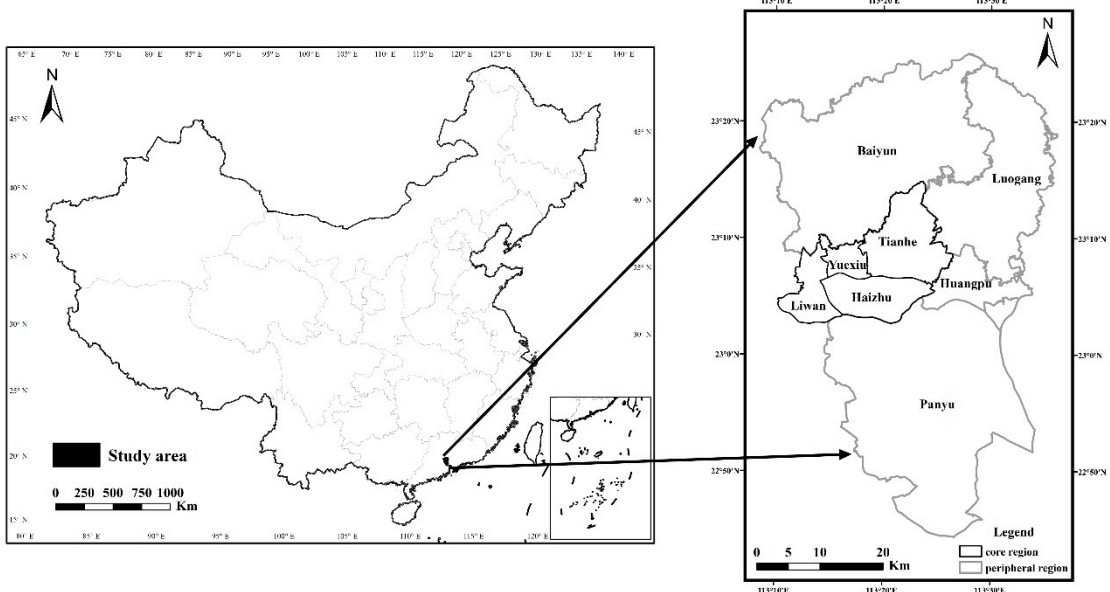

**Figure 1.** Study area and its location in China.

### 2.2. Data and Pre-Processing

#### 2.2.1. Imagery

Free and open datasets including time series Landsat data and yearly ALOS PALSAR data were collected from the United States Geological Survey and the Japan Aerospace Exploration Agency. Landsat imagery included available Landsat Thematic Mapper (TM), Enhanced Thematic Mapper Plus (ETM+), and Operational Land Imager (OLI)/Thermal Infrared Sensor (TIRS) images (path/row 122/044) from 2007 to 2009 and from 2015 to 2017 for identifying urban land change from surface reflectance data; 69 images were obtained over the first three-year period (2007–2009) and 77 images were obtained during the latter three-year period (2015–2017). In addition, we collected Landsat 5 image (path/row 122/044) on 26 July, 2008, and Landsat 8 image (path/row 122/044) on 22 July, 2018, for LST measurements, which included raw data, top-of-atmosphere reflectance data, and surface reflectance data. Before using the optical imagery provided by Landsat TM, Landsat ETM+, and Landsat OLI, images with poor quality were identified and removed: (1) the Fmask algorithm was first applied to develop cloud and cloud shadow layers for time series analysis of Landsat TM, Landsat

ETM+, and Landsat OLI images [38]; and (2) to improve the continuity and usability of the images, we built a scan line corrector (SLC) off layer to mask no-observation pixels in Landsat ETM+ images due to the failure of the on-board SLC from 31 May 2003.

Yearly global 25 m resolution ALOS PALSAR imagery provided L-band dual-polarization data, including horizontal transmit/horizontal receive (HH) and horizontal transmit/vertical receive (HV) polarizations, which are capable of full-polarization and multi-view Earth observation. Here, ALOS PALSAR images from 2007–2009 and from 2015–2017 were collected and subjected to radiation and geometric correction in google earth engine (GEE), which is a free and open platform for batch processing of satellite image data. The PALSAR HH and HV digital number (DN) values were then converted into gamma-naught to reduce the influence caused by the change of the backscattering coefficient in the distance direction by the following calibration coefficient [39]:

$$\gamma^\circ = 10 \times \log_{10}\left\langle DN^2 \right\rangle - 83, \tag{1}$$

To reduce noise, a $3 \times 3$ pixel median filter was applied to PALSAR HH and HV images, which were resampled into 30 m images to match Landsat images by the nearest neighborhood interpolation [40].

### 2.2.2. Ground Reference Data for Approach Training and Validation

We collected 50 completed urban renewal project files from Guangzhou Municipal People's Government during the study period as ground reference data. The specific information in the project files includes the location, scope, duration, transformation types, change in land use type, and so on. According to the urban renewal project files and very high-resolution Google Earth images of the two time periods (approximately 2007 and 2017), we selected 50 urban renewal land (URL) points and 50 general urban land (GUL) points as training data (Figure 2a), which were generated in $60 \times 60$ m$^2$ regions of interest (ROIs).

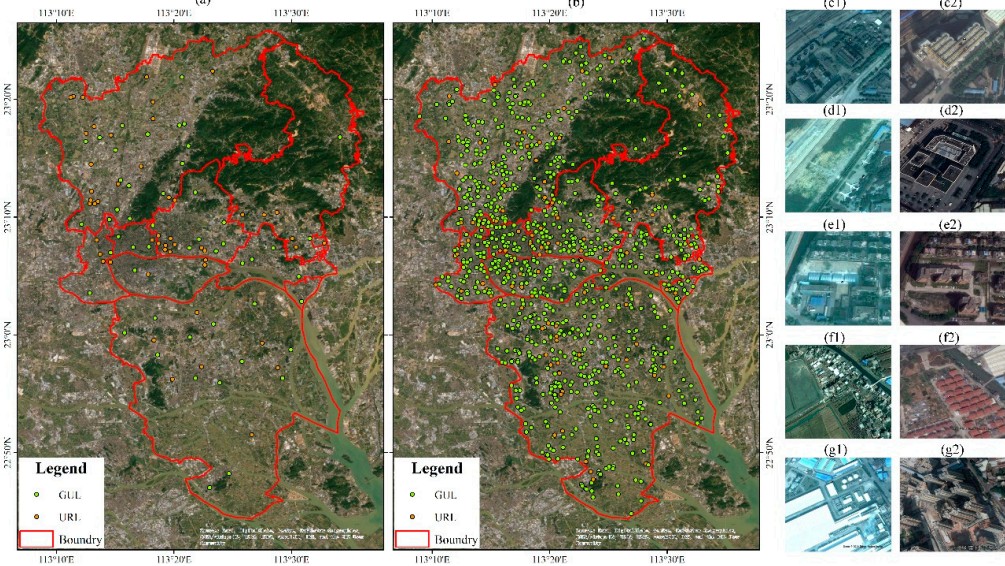

**Figure 2.** Random ground reference for algorithm training (**a**) and resultant accuracy assessment (**b**) of urban renewal land. (**c**–**g**) Very high spatial resolution images of urban renewal in the former and later phase from Google Earth. c (23.0589, 113.5125) is from an old factory (c1, 10/17/2007) and a new factory (c2, 12/5/2016); d (23.1772, 113.2638) shows a change from an airport (d1, 10/9/2007) to an office building (d2, 9/16/2017); e (22.9456, 113.3472) shows a conversion from a factory (e1, 12/30/2006) to an apartment building (e2, 4/1/2017); f (22.9435, 113.4822) is from a village (f1, 2/27/2008) to a house (f2, 3/3/2017); and g (23.1070, 113.5470) is from a factory (g1, 10/17/2007) to an apartment (g2, 1/15/2018).

We also generated 1000 60 m × 60 m rectangles (ROIs) for accuracy assessment through stratified random sampling, with a minimum distance of 1000 m between ROIs (Figure 2b). In total, URL consisted of 100 ROIs, and GUL included 900 ROIs. The ROIs were overlaid with very high spatial resolution images from Google Earth (Figure 2c–g) in the study area for 2007 and 2017, respectively. If more than 50% of the ROIs were URL or GUL, the ROI would be classified as URL or GUL, respectively. Based on these classification results, we constructed a confusion matrix to evaluate the classification accuracy of URL and GUL retrieved from remote sensing images, including producer accuracy, user accuracy, overall accuracy, and the kappa coefficient.

### 2.3. Methodology

We collected time series Landsat TM, ETM+, OLI images, and ALOS PALSAR images and carried out data preprocessing (Figure 3). Then, urban land in 2007 and 2017 was identified using annual maximum normalized difference vegetation index (NDVI) calculated by Landsat imagery and HH gamma-naught backscatter data from ALOS PALSAR. We identified urban renewal land using HV gamma-naught backscatter data and 100 ROIs from urban renewal project files and Google Earth images in non-change of urban land. And urban renewal land was validated by 1000 ROIs randomly sampled from Google Earth images. After retrieving the LST from Landsat imagery, we further analyzed the influence of urban renewal on LST.

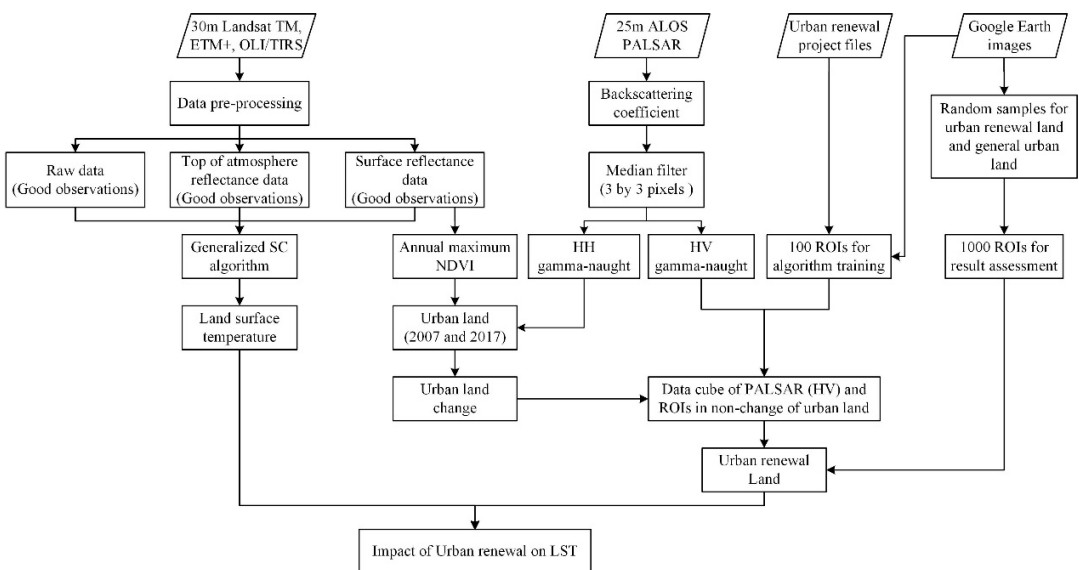

**Figure 3.** Workflow for analyzing the impact of urban renewal on land surface temperature (LST), based on the combination of Landsat Thematic Mapper (TM)/Enhanced Thematic Mapper Plus (ETM+)/Operational Land Imager (OLI)/Thermal Infrared Sensor (TIRS) and Advanced Land Observation Satellite (ALOS) Phased Array type L-band Synthetic Aperture Radar (PALSAR) images.

#### 2.3.1. Identification of Urban Land and Urban Change

Generally, urban land is composed mainly of three-dimensional buildings, mixed with different land cover types (e.g., roads, grasslands, trees, and water bodies). The greenness is relatively low inside urban land areas. To identify urban land from other land use types, e.g., croplands, forests, and wetlands, the NDVI calculated using Landsat imagery and HH gamma-naught backscatter data from ALOS PALSAR were jointly adopted in this study to map urban land, based on earlier successful attempts using this approach [40]. The NDVI is given by

$$\text{NDVI} = \frac{\rho_{\text{nir}} - \rho_{\text{red}}}{\rho_{\text{nir}} - \rho_{\text{red}}}, \tag{2}$$

where $\rho_{nir}$ and $\rho_{red}$ are the land surface reflectance values of red (0.63–0.69 μm) and near-infrared (NIR; 0.77–0.90 μm) bands for Landsat TM and Landsat OLI/TIRS images. We calculated the maximum NDVI value ($NDVI_{max}$) for each year of the study (2007, 2008, 2009, 2015, 2016, and 2017).

PALSAR HH and Landsat $NDVI_{max}$ represent the features of building structure and greenness, respectively [40]. Therefore, we applied both of them to map urban land in the study area from 2007–2009 and from 2015–2017. Urban land over the six years represented in the two time periods was extracted by relatively high HH backscatter values and relatively low $NDVI_{max}$ values according to the mean value and standard deviation of land cover types from Landsat and PALSAR images satisfying the threshold referring to [40]:

$$(HH \geq -9) \text{ \&\& } (NDVI_{max} \leq 0.6), \tag{3}$$

To obtain the urban land distribution in 2007 and 2017, and to reduce the possibility of misjudgment of urban land, we classified pixels as urban if they were present for more than 2 years of either of the study periods from 2007–2009 and 2015–2017, i.e., 2 of the 3 years of the individual study period. For example, each pixel of the three years from 2007–2009 or from 2015–2017 had eight urban land (U) and non-urban land (N) permutations (UUU, NUU, UNU, UUN, NNU, NUN, UNN, and NNN); unreasonable permutations were filtered to retain consistency (UNU→UUU, NUN→NNN). After mapping urban land and non-urban land in 2007 and 2017, we calculated four types of land use changes in the study area (Figure 4): urban land to urban land (UU), urban land to non-urban land (UN), non-urban land to urban land (NU), and non-urban land to non-urban land (NN).

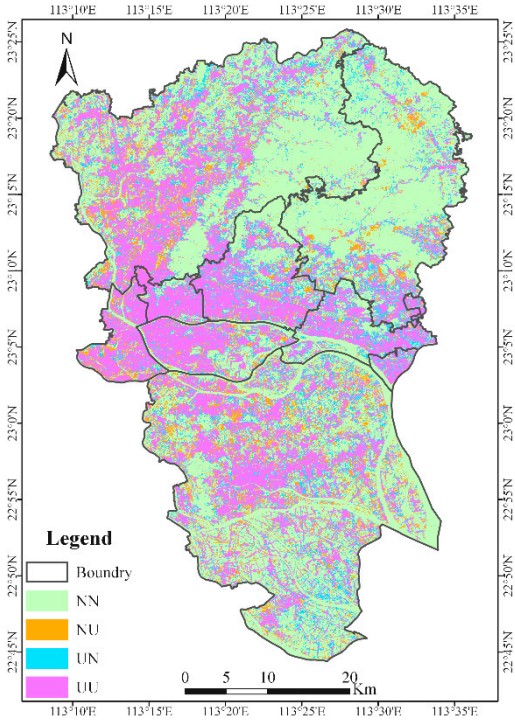

**Figure 4.** Different types of land use changes in the study area from 2007 to 2017.

### 2.3.2. Algorithms for Identifying Urban Renewal Land through PALSAR Images

In general, because the floors and heights of buildings commonly show considerable change before and after the urban renewal process, this study extracted URL on a large scale according to the height difference of buildings between the former period and the latter period. Previous research demonstrated the sensitivity of synthetic aperture radar (SAR) to building height [41,42].

Therefore, we identified URL from UU based on the building height retrieval from PALSAR imagery in different periods.

Figure 5 showed two-dimensional scatter plots of the HH and HV average values from 2007–2009 and from 2015–2017. URL could easily be distinguished from GUL in the range in which the absolute value of the difference between HV average value during 2007–2009 and 2015–2017 was greater than 8, where the URL points held a relatively compact distribution pattern. Although URL could be distinguished from GUL in the range where the absolute value of the difference between the HH average value in 2007–2009 and 2015–2017 was greater than 6.5, some GUL points were misjudged as URL, and vice versa. The URL points held a relatively discrete distribution pattern, thus the average value of PALSAR HV from different time periods was chosen to further map URL in the study area.

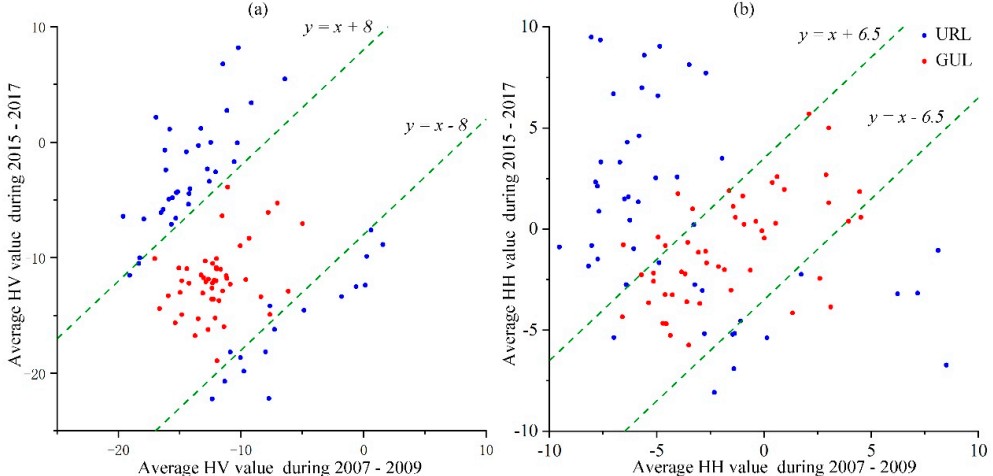

**Figure 5.** Two-dimensional scatter plots of (**a**) average horizontal transmit/vertical receive (HV) gamma-naught and (**b**) average horizontal transmit/horizontal receive (HH) gamma-naught from 2007–2009 and 2015–2017 in the study area.

As one of the land use change types of UU, URL was mainly featured by the differences in PALSAR HV backscatter average values before and after urban renewal in our study. To distinguish URL from GUL, we used the following threshold value of the difference between the two periods:

$$|HV_{former} - HV_{latter}| > 8,　　　　　　　　　　　(4)$$

where $HV_{former}$ is the average HV value of each pixel of UU from 2007 to 2009, and $HV_{latter}$ is the average HV value of the corresponding pixel from 2015 to 2017. Because some pixels of UU were mixed with URL and GUL, the lowest 5% values of ground reference for URL were excluded to reduce false identification and confusion error. Moreover, we removed the pixels that covered less than four independent pixels in the range of $3 \times 3$ pixels, to reduce URL noise.

### 2.3.3. Land Surface Temperature Retrieval

Urban renewal aims to replace functionally decaying urban elements with brand-new urban functions by reallocating land use and increasing urban greenness, such as the replacement of heavily polluted factories and thermal power stations with commercial and residential areas. As such, urban renewal was expected to have a significant effect on LST and UHI.

Thermal infrared sensor (TIRS) data from remote sensing is considered to be an effective way to retrieve the LST. We selected two scenes from Landsat images to retrieve the LST; the Landsat Thematic Mapper (TM) on 26 July 2008, and the Landsat Thermal Infrared Sensor (TIRS) on 22 July 2018. The previous image had no cloud coverage in the study area, whereas the latter image had 14%

cloud coverage, corresponding to the image with the least amount of cloud coverage of Landsat images in the summer of 2015–2018. Given that the main coverage area of cloud was farmland and woodland, cloud coverage had little impact on this research. Specifically, the LST of the study area was retrieved from Band 6 of the Landsat TM imagery on 26 July 2008, and from Band 10 of Landsat TIRS imagery on 22 July 2018, using the generalized single-channel algorithm [43,44], which has been proven an effective method to retrieve LST from Landsat imagery [45–47]. This algorithm mainly calculates LST from a combination of the surface emissivity, at-sensor registered radiance, atmospheric functions, and parameters dependent on Planck's function.

Figure 6 showed the LST outcomes retrieved from Landsat TM imagery on 26 July 2008, and Landsat TIRS imagery on 22 July 2018. The overall temperature in the latter period was much higher than that in the former period. Compared with the former period, the relatively high temperature areas showed an obvious outward diffusion trend in the latter period.

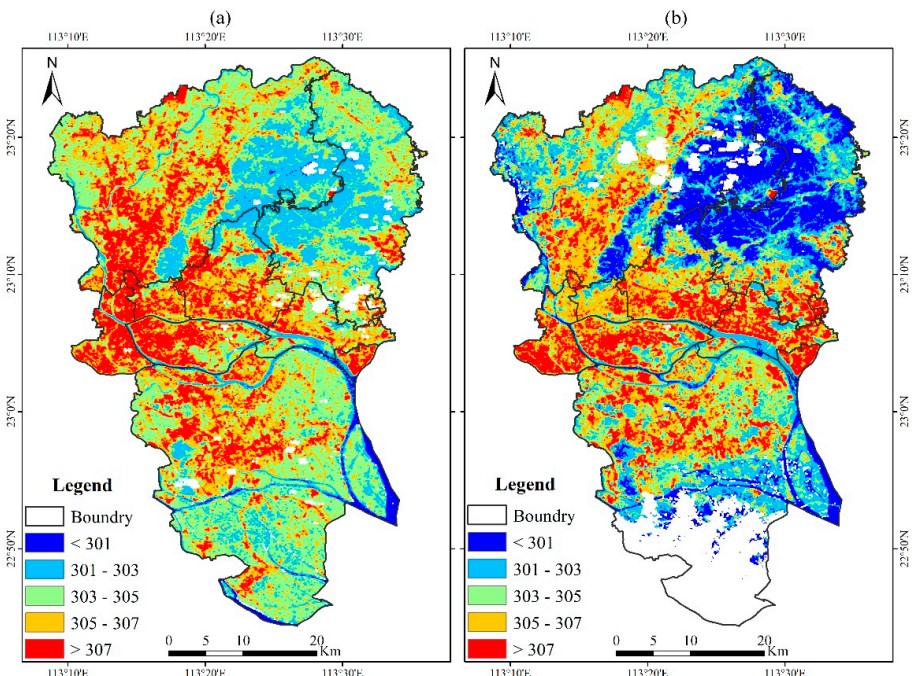

**Figure 6.** Spatial distribution of LST (Kelvin) on 26 July 2008 (**a**) and on 22 July 2018 (**b**) in the study area.

### 2.3.4. Spatial Statistics

To reflect the difference in heat generation between URL and GRL in different periods, we calculated the LST difference between the two types of urban land:

$$D_{i,j} = T_{i,j}^{URL} - T_{i,j}^{GUL}, \tag{5}$$

where $T_{i,j}^{URL}$ is the mean LST of URL in district j during period i; $T_{i,j}^{GUL}$ is the mean LST of GUL in district j during period i; and $D_{i,j}$ is the mean LST difference between URL and GUL in district j during period i. If $D_{i,j} < 0$, the mean LST of the URL is lower than those of the GUL. Otherwise, the mean LST of the URL is higher than those of GUL.

To further clarify the impact of urban renewal on LST in urban areas, we first calculated the mean LST difference between URL and GUL in different periods, and then calculated the difference between the two differences to show the specific effects of urban renewal on different urban areas:

$$D_j = D_{i+1,j} - D_{i,j}, \tag{6}$$

where $D_{i+1,j}$ is the mean LST difference between URL and GUL in district j during period i + 1, and $D_j$ is the difference between the latter period and the former period. If $D_j < 0$, urban renewal reduced the LST in specific urban areas. Otherwise, urban renewal led to an increase in LST.

## 3. Results

### 3.1. Precision of the Urban Renewal Interpretation

Based on the ROIs of URL and GUL judged in Section 2.3, we assessed the precision of assigning URL and GUL using the confusion matrix (Table 1). The confusion matrix showed that both URL and GUL identifications were assessed with high accuracy. The overall accuracy and kappa coefficient of URL were 97% and 0.82, respectively. The error classification of URL may be caused by the complex land cover types and fragmented landscape patterns of the pixels. The user accuracy and producer accuracy of GUL were up to 98%, which was higher than for URL.

**Table 1.** Accuracy assessment of urban renewal land and general urban land using the ground reference data from Google Earth.

| Land Use Type | | Ground Reference Data | | User Accuracy | Overall Accuracy | Kappa Coefficient |
|---|---|---|---|---|---|---|
| | | GUL | URL | | | |
| Classified results | GUL | 886 | 18 | 98% | | |
| | URL | 14 | 82 | 85% | 97% | 0.82 |
| Producer accuracy | | 98% | 82% | | | |

### 3.2. Spatial Distribution Characteristics of Urban Renewal Land

During 2007–2017, the URL area was 13.18 km$^2$ in total, accounting for 1.88% of the area of urban land (Table 2). URL was attached to GUL and widely distributed in the study area (Figure 7); additionally, URL showed a relatively concentrated distribution in the center of the study area. For the core region, the URL proportion reached 2.49%, which was 0.61 higher than the overall level. The URL proportions for Tianhe and Haizhu districts were relatively high among the core region, at 2.93% and 2.46% respectively. For the peripheral region, the URL proportion was 1.67%, which was 0.21 lower than the overall level. Although the areas of URL in Panyu and Baiyun districts were relatively high, accounting for 45.19% of the total URL, the proportions of URL in Luogang and Huangpu districts were relatively high among the peripheral region.

**Table 2.** Spatial distribution of urban renewal land (URL) and general urban land (GUL) in different districts from 2007–2017.

| Region | Districts | Urban land (km$^2$) | URL (km$^2$) | Proportion (%) | GUL (km$^2$) | Proportion (%) |
|---|---|---|---|---|---|---|
| Core region | Liwan | 42.46 | 0.86 | 2.03 | 41.60 | 97.97 |
| | Yuexiu | 24.86 | 0.55 | 2.19 | 24.31 | 97.81 |
| | Haizhu | 49.46 | 1.22 | 2.46 | 48.25 | 97.54 |
| | Tianhe | 65.17 | 1.91 | 2.93 | 63.26 | 97.07 |
| Peripheral region | Baiyun | 196.23 | 2.79 | 1.42 | 193.44 | 98.58 |
| | Huangpu | 41.02 | 0.80 | 1.94 | 40.22 | 98.06 |
| | Panyu | 225.97 | 3.16 | 1.40 | 222.81 | 98.60 |
| | Luogang | 55.45 | 1.89 | 3.42 | 53.55 | 96.58 |
| | total | 700.63 | 13.18 | 1.88 | 687.45 | 98.12 |

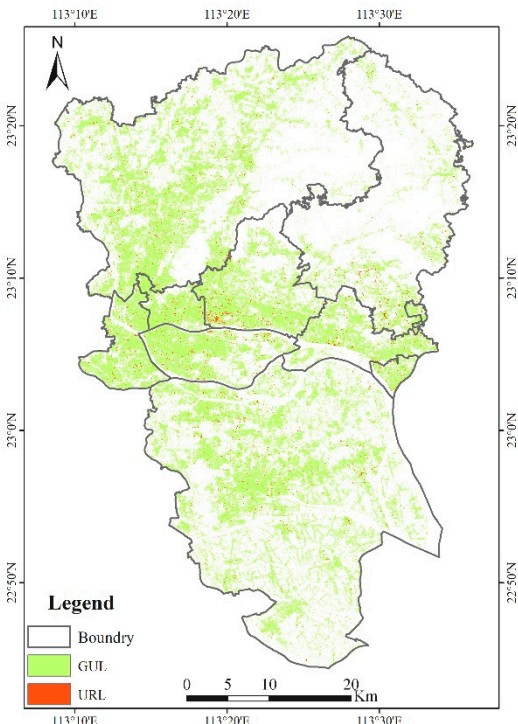

**Figure 7.** Spatial distribution of urban renewal land (URL) and general urban land (GUL) in the study area.

### 3.3. Spatiotemporal Change in the Land Surface Temperature

The LST differences between URL and GUL in different regions were used to indicate the thermal difference between the two types of urban land in different periods (i.e., $D_{2007}$ and $D_{2017}$ in Table 3). The actual impact of urban renewal processes on LST was obtained by calculating the difference in the LST over different periods (i.e., D in Table 3). After urban renewal, the relative LST of Liwan, Yuexiu, Haizhu, and Tianhe districts dropped by 0.88, 0.42, 0.43, and 0.10 K, respectively. Thus, the urban renewal process in the core region was conducive to the decline in LST of urban areas. However, Baiyun, Huangpu, Panyu, and Luogang districts in the peripheral region presented completely opposite characteristics, with a rise in the LST of 0.98, 1.03, 1.63, and 2.11 K.

**Table 3.** Land surface temperature (LST, units of Kelvin) difference of GUL and URL in different regions from 2007–2017.

| Region | Districts | 2007 | | | 2017 | | | D |
|---|---|---|---|---|---|---|---|---|
| | | GUL | URL | $D_{2007}$ | GUL | URL | $D_{2017}$ | |
| Core region | Liwan | 307.49 | 307.47 | −0.02 | 307.21 | 306.31 | −0.90 | −0.88 |
| | Yuexiu | 306.95 | 306.90 | −0.04 | 306.28 | 305.81 | −0.46 | −0.42 |
| | Haizhu | 307.25 | 307.26 | 0.01 | 307.28 | 306.86 | −0.42 | −0.43 |
| | Tianhe | 306.87 | 307.03 | 0.15 | 306.72 | 306.77 | 0.05 | −0.10 |
| Peripheral region | Baiyun | 306.72 | 306.84 | 0.12 | 305.66 | 306.75 | 1.10 | 0.98 |
| | Huangpu | 306.23 | 306.28 | 0.05 | 307.44 | 308.53 | 1.08 | 1.03 |
| | Panyu | 306.23 | 306.39 | 0.15 | 305.73 | 307.51 | 1.78 | 1.63 |
| | Luogang | 305.98 | 305.85 | −0.13 | 305.93 | 307.91 | 1.97 | 2.11 |

The urban renewal process had the opposite thermal effect on the districts in the core and peripheral regions of Guangzhou from 2007 to 2017. Specifically, urban renewal in areas with relatively high populations and land urbanization (e.g., Liwan, Yuexiu, Haizhu, and Tianhe districts) was beneficial in reducing the LST. In contrast, urban renewal in areas with a relatively low population

and minimal land urbanization, such as Baiyun, Huangpu, Panyu, and Luogang districts, showed an increase in LST with urban renewal. This phenomenon can be explained by the following two factors. On the one hand, because the initial building density and population density in the core region were already very high, urban renewal had little impact on the building density and population density in that area. The urban renewal process effectively added green landscape, which was conducive to adjusting the urban microclimate and alleviating urban thermal environmental risk [48]. At the same time, most of the target sites for urban renewal were inefficient sites, such as urban villages and old factories and thermal power stations, which directly increased the environmental and thermal threats, compared with other urban land types. Thus, urban renewal alleviated these problems through the transformation of extensive land inefficiencies. On the other hand, the initial building density of edge areas was relatively small in the peripheral region; however, the building density and population increased significantly after urban renewal, which resulted in corresponding changes in the urban microclimate and thermal environment. Some related studies have also pointed out that the increase in the building and population densities had a significant impact on UHI and LST [49,50]. Therefore, the urban renewal process in peripheral region districts promoted higher LSTs, to varying intensities.

## 4. Discussion

### 4.1. Comparison with Previous Studies

A large number of studies have shown the close relationship between urbanization and the change in urban thermal environments, as evidenced by higher LST [15,51–53]. But what is the actual impact of urban renewal on LST? The main objective of our research was to identify the key mechanism of urban renewal on LST and to analyze further whether the impact differed with location and/or degree of urbanization. Our research results revealed a difference in the impact of urban renewal on the LST between core and peripheral districts. Specifically, urban renewal in the core region with a relatively high intensity of urbanization was beneficial to reducing LST, whereas in the peripheral region, urban renewal increased the LST.

Despite there being few studies in this field, some useful exploration has been carried out. Hou et al. showed that the LST in several high-temperature areas of central Fuzhou City declined with the development of urban renewal from 2003 to 2016, which is consistent with the changing tendencies of districts in the core region of our study area [32]. Pan et al. examined the Tianhe District of Guangzhou City as a key research area, confirming that changes in land use and spatial structures associated with urban renewal significantly eliminated or weakened the intensity of UHIs [33]. Peng et al. presented a comprehensive mathematical model describing fluid flow and heat transfer characteristics to provide strategies for urban renewal and improving the wind and thermal environments in the old city district of Wuhan, which has a dense population and experiences a strong UHI effect [34]. Existing research has focused on the impact of urban renewal on LST in areas with high urbanization intensity, high populations, and high building densities. However, no prior research has revealed the spatial difference of urban renewal impact on LST, especially in the context of a comparative analysis of the impact difference of areas with different urbanization degrees. Therefore, for regions with relatively low urbanization, such as urban peripheral regions, the impact of urban renewal on LST deserves further study.

### 4.2. Future Work

With the continuous progress of urbanization, the urban landscape continues to evolve [54,55]. Many sites and buildings that are not suitable for the development of modern cities, such as old towns, factories, and villages, must be updated. Thus, with increasing urban populations, there has been great interest in urban renewal and the preservation of resources for a sustainable and healthy living environment [32,34]. Given the lack of extensive research in this area, we hope that the results from this study will provide a framework for future research on LST changes caused by urban renewal

processes. Our research group hopes to improve on the spatiotemporal acquisition accuracy of imagery from large-scale urban renewal areas and land type identification for LST calculations. For example, some houses in urban villages have illegally-built additional floors, resulting in obvious changes in building height. Although illegal construction does not belong to the category of urban renewal, it may be identified as URL. In addition, specific landscape changes in the urban renewal process with LST changes can be used to refine the heat exchange processes of urban microclimates driven by different landscape change types, so as to provide more detailed guidance for future urban renewal projects.

## 5. Conclusions

Urban renewal is an effective method for solving shortages in urban land resources with urbanization by redeveloping land that is used inefficiently. At the same time, urban renewal affects the urban microclimate and is steadily receiving more attention. In this study, we proposed a general, flexible framework to determine how urban renewal processes influence LST, using multi-source remote sensing images. We presented a case study of core and peripheral regions in Guangzhou City, China, between 2007 and 2017 to analyze the spatial differences in different urbanized areas on the basis of urban renewal area identification and LST remote sensing inversion. Our results indicated that the urban renewal process is conducive to a decline in LST in core region districts with relatively high urbanization, whereas peripheral regions with relatively low urbanization presented the opposite characteristics. The outcomes can be explained by the joint influence of population density, building density, and landscape pattern changes in the process of urban renewal. It is our hope that the results from this study provide a new perspective on the impact of urban renewal on urban microclimates and thermal environmental changes to promote greater awareness of this issue in future planning and management practices.

**Author Contributions:** Data analysis and interpretation was conducted primarily by Z.Q. and L.L., with contributions from Y.Q., X.X., B.W., and Z.L.; Z.Q. wrote the paper and edited by Z.L.; research was directed by Z.L. All authors have read and agreed to the published version of the manuscript.

**Funding:** This research was funded by National Natural Science Foundation of China (Grant #: 41601082, 41971389, 41501472) and the Major Projects of High Resolution Earth Observation Systems of National Science and Technology (Grant #: 05-Y30B01-9001-19/20-4).

**Conflicts of Interest:** The authors declare no conflict of interest.

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
