# Peer review of "The Impact of Urban Renewal on Land Surface Temperature Changes: A Case Study in the Main City of Guangzhou, China"

_remotesensing, doi:10.3390/rs12050794_

Round 1

Reviewer 1 Report

Dear authors,

After a very careful rading of your work entitled "The impact of urban renewal on land surface temperture changes: a case study ..." I have found a very well done work, well presented and organized, clear in concepts and methodology. The methodology and data processing is very good even when combining data from different sensors with different ranges and spatial resolutions.

Section 2.3 (methodology) must be better explained

Good luck

Reviewer 2 Report

Dear Authors,

The topic and context attract attention for many readers from various disciplines. The study is worth to be published in Remote Sensing after conducting the revisions. Please consider the attached file for my comments.

All the best.

Reviewer 3 Report

The article "The Impact of Urban Renewal on Land Surface Temperature Changes: A Case Study in the Main City of Guangzhou, China" meets the thematic requirements of "satellite remote sensing of urban thermal environment: advances, challenges, and opportunities." It is of innovative and practical value to study how urban renewal process influence the urban thermal environment in different regions. The authors did a good experiment and the outcomes are also very interesting and inspiring.

However, there are some questions in the article that need to be revised subsequently:

(1)Why do the authors choose 2007 and 2017 as time nodes? Is it because urban renewal and LST changes in the region of interest can be better explained within this time period?

(2)The urban renewal interpretation showed high precision in Section 3.1. But In P10, L277-278, it showed that “the overall accuracy and kappa coefficient of URL were 97% and 0.82, respectively”. Is this the precision for a specific presentation URL? Or is it accuracy including GUL? Please check whether the writing is wrong.

(3)Paper’s English needs to be further improved and edited (sentence structure and vocabulary). I strongly suggest that you obtain assistance from a colleague who is well-versed in English or whose native language is English.

(4)In P3, L61-67. “For example, based on Aqua/Terra…” Restructure and make the sentence shorter.

(5)Please check the format of some details carefully. For example, in P3 L108, P3 L110, in P4, L148, the writing styles of square meter and square kilometer are not standard.

(6)In P3, L239 and L241, The use of short names for thermal sensors should be unified.

(7)Table 3 in Page 11 spans two pages. The same table should be represented on a single page. At the same time, the size of the figures should be consistent, such as Figure 4 and Figure 7.

Please check the format again in the revised version. It is suggested to receive the paper after modification.

Round 2

Reviewer 1 Report

Congratulations for a nice work!!!

Reviewer 2 Report

Dear Authors,

Your revisions on the paper are appreciated. Now, the paper can be published as it is.

All the best.

Reviewer 3 Report

Dear Authors,

I think the current version of paper is acceptable now, even though there are two minor issues. Authors can revise the two during the future proofreading.

Line 46-47, reverse the order to present a right 'cause and effect'.

Line 55-56, remove it. It is repetitive to previous contents.

Overall, I suggest the acceptance of this paper.